# OpenReview forum: "Revisiting Multi-Agent Debate as Test-Time Scaling: When Does Multi-Agent Help?"
_ICLR.cc/2026/Conference — Submitted to ICLR 2026_

### Official Review · Reviewer_yLiU · 2025-10-29

**Soundness:** 2
**Presentation:** 3
**Contribution:** 2
**Rating:** 4
**Confidence:** 4

**Summary:**

This paper reframes multi-agent debate (MAD) as a test-time scaling strategy. The authors evaluate homogeneous and heterogeneous MAD on mathematical reasoning and safety. For math, MAD rarely beats strong self-consistency, with slight gains only on the hardest problems; for safety, adding agents and diversity generally lowers attack success via collaborative refinement, with benefits most pronounced under heterogeneous configurations.

**Strengths:**

- Clear conceptual positioning of MAD as test-time scaling with two distinct features: collaborative refinement and diverse exploration.


- Systematic, controlled comparisons against strong self-agent baselines across tasks, difficulties, model sizes, and diversity settings, with matched generation budgets.
- Provide actionable insights for math and safety tasks.

**Weaknesses:**

The primary concern is the generalizability of the conclusions, both within and across domains. Although the paper reports task-specific findings, there is no clear trend across tasks, and the divergent conclusions for math vs. safety make it difficult to infer whether MAD will help in other settings. Moreover, evaluating only a vanilla MAD configuration does not preclude gains from alternative debate designs. Prior work, e.g., Encouraging Divergent Thinking in Large Language Models through Multi-Agent Debate, has reported improvements on math, and other evaluations (e.g., Should We Be Going MAD? A Look at Multi-Agent Debate Strategies for LLMs) are cited but not compared in depth. Overall, the lack of thorough cross-study comparison and limited protocol coverage raises questions about both external validity and novelty.


- Primary focus on math reasoning and safety; it’s unclear how well findings generalize to other domains (planning, coding, long-form generation) or to alternative debate protocols beyond the vanilla setup.
- Safety evaluation relies on a closed-source judge (gpt-4o-mini), which can introduce evaluation bias; robustness to the choice of judge is not established.
- The framing is largely confined to multi-agent debate; there’s limited comparison against other multi-agent collaboration paradigms, making it unclear whether conclusions transfer beyond debate. Is it limited only to a multi-agent debate framework? Can it be generated for other Multi-agent collaboration systems?

**Questions:**

- Even within math and safety, how robust are the conclusions across subdomains and difficulty regimes?
- Where do MAD’s gains primarily come from—multi-agent diversity, the debate/refinement phase, or simply increased sampling? Can you decompose the contribution of each?




- Do you have guidance or a heuristic for deciding when to use MAD on a new task?
-  Why use majority voting over two sampled generations, and how are disagreements/ties resolved?
- How should practitioners choose the number of agents? Any compute-aware heuristic or scaling law for picking an effective agent count?

---

> ### Author Response · Authors · 2025-11-21
> **Response to Reviewer yLiU (1/2)**
>
> Thank you so much for providing helpful feedback. We will address your concerns one by one. We have simplified your comments for easier reference and included our respective responses.
>
> ---
>
> ### **W1: Divergent Conclusions**
> > The primary concern is the generalizability of the conclusions... there is no clear trend across tasks, and the divergent conclusions for math vs. safety make it difficult to infer whether MAD will help in other settings... evaluating only a vanilla MAD configuration does not preclude gains from alternative debate designs...
>
> **Response:**
> We want to clarify that our "divergent conclusions" across tasks are not a limitation, but rather the core strength of our **unified framework**.
> * **Unified Mechanism:** Our work unifies these observations by explaining *why* MAD behaves differently. We show that the effectiveness of MAD relies on *Diverse Exploration* and *Collaborative Refinement*. The "divergence" arises because different task types (Solution-Finding vs. Response-Judgment) benefit differently from these mechanisms. This allows us to predict *where* MAD will be meaningful.
> * **Choice of Vanilla MAD:** We deliberately focused on **Vanilla MAD** because it is the most scalable, generalizable, and fundamental form of debate. Complex designs (like assigning specific Pro/Con/Judge personas [1] or complex prompting strategies [2]) often introduce confounding variables and limit scalability. By analyzing the fundamental debate structure, our findings remain applicable across various implementations.
> * **External Validity:** As detailed in our **General Response** and updated **Related Work**, we have extensively compared our findings with other studies. Unlike works that require specific role-play (limiting generalization), our framework explains the fundamental interaction dynamics.
>
> ---
>
> ### **W2: Limited Tasks**
> > Primary focus on math reasoning and safety; it’s unclear how well findings generalize to other domains...
>
> **Response:**
> We agree that broader coverage is important. As mentioned in our **General Response**, we have added **GPQA-Diamond** (to validate findings in Solution-Finding) and **RewardBench** (to validate findings in Response-Judgment). The results on these new tasks align perfectly with our original conclusions, demonstrating that our framework generalizes beyond just Math and Safety.
>
> ---
>
> ### **W3: Closed Judge**
> > Safety evaluation relies on a closed-source judge (gpt-4o-mini)... robustness to the choice of judge is not established.
>
> **Response:**
> To verify the reliability of our evaluation, we conducted cross-validation experiments using **Gemini-2.5-Pro**, a significantly stronger model than GPT-4o-mini.
>
> The results using the stronger judge show the exact same trends as our original evaluation. This confirms that the observed effectiveness of MAD in safety tasks is a robust phenomenon and not an artifact of a weak evaluator.
>
> These validation results are included in ${\color{blue}\text{Appendix H.5}}$.
>
> ---
>
> ### **W4: Different Multi-Agent Paradigms**
> > The framing is largely confined to multi-agent debate... limited comparison against other multi-agent collaboration paradigms...
>
> **Response:**
> As indicated by our title, our primary focus is strictly **Multi-Agent Debate (MAD)**. We prioritize MAD because it is highly scalable (parallelizable) and possesses distinct, analyzable features (consensus-seeking). In contrast, other multi-agent forms—such as complex agent orchestration or workflow engineering—are considered out of scope for this study. These approaches involve an excessive number of design variables (e.g., workflow topology, tool use), which hinders the distinct, fundamental analysis we aim to provide.

---

> ### Author Response · Authors · 2025-11-21
> **Response to Reviewer yLiU (2/2)**
>
> ### **Q1: Subdomains and Difficulty Regimes**
> > Even within math and safety, how robust are the conclusions across subdomains and difficulty regimes?
>
> **Response:**
> This is an excellent question. We have indeed analyzed performance across difficulty regimes, particularly for mathematical reasoning (${\color{blue}\text{Section 4.1, Page 5}}$).
> * **Difficulty Correlation:** We find that as question difficulty increases, the potential for MAD to be effective increases slightly.
> * **Quantitative Evidence:** On the MATH500 dataset, compared to Self-Consistency, MAD’s gains with 4-8 agents are only positive for the hardest tier (Level 5). Specifically, we see a gain of 3.2% (4 agents) and 1.7% (9 agents) on Level 5, whereas gains are negligible or negative on Levels 1-4.
> * **Takeaway:** While difficulty does modulate effectiveness, the *task type* remains the dominant factor.
>
> ---
>
> ### **Q2: Practical Guidance**
> > Where do MAD’s gains primarily come from... Can you decompose the contribution of each?
>
> **Response:**
> As outlined in our **Discussion & Conclusion**, our practical guidance is derived from decomposing the gains into *diverse exploration* vs. *collaborative refinement*:
> 1.  **For Response-Judgment Tasks:** Use **Heterogeneous MAD**. The gains come from both diverse exploration (covering the search space of errors) and refinement (consensus).
> 2.  **For Solution-Finding Tasks:** Use **Single-Agent Scaling** (Self-Consistency). MAD acts primarily as compute scaling here. While MAD *could* theoretically help if meaningful diverse solutions are guaranteed, current models struggle to maintain diversity without hallucinating, making simple scaling more efficient.
>
> ---
>
> ### **Q3: Answer Aggregation**
> > Why use majority voting over two sampled generations, and how are disagreements/ties resolved?
>
> **Response:**
> We have added the experimental setup details for clarification.
> * **Reasoning Tasks:** We resolve ties by defaulting to **Agent 0**. We evaluated random selection for tie-breaking but observed no significant performance difference. In our heterogeneous model settings, Agent 0 is consistently designated as the strongest model (Qwen).
> * **Safety Tasks:** We employ a **judge model** to definitively resolve ties or evaluate the final consensus.
>
> ---
>
> ### **Q4: Practical Guidance on the number of agents**
> > How should practitioners choose the number of agents? Any compute-aware heuristic...?
>
> **Response:**
> The number of agents plays a significant role in facilitating parallel exploration.
> * **Diminishing Returns:** Empirically, we find that while increasing the count from two to three agents yields noticeable performance gains, adding agents beyond this point results in diminishing returns.
> * **Hardware-Aware Strategy:** Since agents can be parallelized across distinct GPUs, practitioners should align the number of agents with their available computational resources. We recommend **three agents** as the optimal heuristic, balancing performance gains with the cost of parallel instantiation.

---

> ### Author Response · Authors · 2025-11-27
>
> Dear Reviewer yLiU,
>
> We truly appreciate your time and effort in reviewing our work.
>
> As the discussion period is nearing its end, we kindly remind you that only a few days remain for further comments or questions.
>
> In response to your feedback, we have provided a detailed response and added a summary of our response recently.
>
> We kindly ask if you have any additional concerns or questions that we may address during the remaining discussion period.
>
> Thank you once again for your valuable insights.
>
> Best regards,
> The Authors

---

### Official Review · Reviewer_2Jxw · 2025-10-31

**Soundness:** 3
**Presentation:** 3
**Contribution:** 2
**Rating:** 4
**Confidence:** 4

**Summary:**

This manuscript reframes Multi‑Agent Debate (MAD) as a test‑time scaling mechanism that combines diverse parallel exploration with sequential collaborative refinement. Under matched generation budgets, the study compares MAD to strong self‑agent baselines: Self‑Consistency (SC) and Self‑Refinement (SR), across mathematical reasoning and safety tasks. The main findings are: (i) in math, SC typically outperforms or matches MAD, with only slight gains for MAD on very difficult problems where collaborative refinement can validate a rare correct trajectory; (ii) in safety, SR can amplify attack success rates (ASR) for smaller models, whereas MAD with more agents and heterogeneous configurations tends to reduce ASR and sometimes matches/surpasses the safest single agent as rounds increase.

**Strengths:**

1. The paper conceptualizes Multi-Agent Debate (MAD) as a test-time scaling strategy, clearly analyzing its effectiveness along two distinct dimensions: parallel scaling to enhance generative diversity and sequential refinement for iterative critique and revision. This approach effectively elucidates MAD’s relative advantages over single-agent baselines.

2. The authors systematically evaluate and compare multi-agent and single-agent approaches across different tasks, incorporating diverse agent configurations through variations in models and prompts. This comprehensive analysis provides nuanced insights into how heterogeneity influences multi-agent outcomes.

3. The employment of detailed diagnostics (e.g., Best-of-Correctness [BoC], Best-on-Follow [BoF]) and the thorough analysis of answer transitions from initial to final states further clarify the internal dynamics underpinning the effectiveness of multi-agent strategies.

**Weaknesses:**

1.The study categorizes tasks into two broad regimes—those requiring extensive exploration of large search spaces and those necessitating consensus among limited options. However, for each category, the authors select only one specific type of task (mathematics and safety, respectively), which limits the representativeness and generalizability of the conclusions.

2.Prior research ([1]) appears to have explored similar comparative analyses between MAD and Self-Consistency (SC), including investigations into heterogeneity-driven MAD approaches, yet no citation or discussion of this work is provided in the manuscript.

3.The reliance solely on GPT-4o-mini for evaluating the Attack Success Rate (ASR) in safety tasks raises concerns regarding evaluator credibility. Given that GPT-4o-mini itself may face limitations in handling complex safety evaluations, incorporating human validation or cross-validation with more robust evaluators would significantly enhance confidence in these results.

4. While alignment across methods is established based on the number of generated responses, MAD methods incur additional overhead in context tokens due to cross-agent interactions and extended prompts. This aspect is not quantified, limiting the fairness and practical interpretability of the computational cost comparisons.

5. Figure 7 inaccurately labels metrics as "accuracy" rather than ASR, creating confusion and reducing clarity. Consistency across the manuscript's terminology and visual presentations should be ensured.

[1] Zhang, Hangfan, et al. "Stop Overvaluing Multi-Agent Debate—We Must Rethink Evaluation and Embrace Model Heterogeneity." arXiv preprint arXiv:2502.08788 (2025).

**Questions:**

1.Could the authors validate their proposed two-regime task classification with additional datasets (e.g., code generation, knowledge-intensive QA, or tool-assisted planning) to better substantiate the generalizability of their conclusions?

2.Can the authors provide evidence regarding the reliability and accuracy of GPT-4o-mini as the evaluator for safety-related tasks? For instance, could human evaluators or comparisons with stronger models be included, and consistency or inter-rater agreement explicitly reported?

3.Could the authors provide a detailed comparison of token usage and computational resources (e.g., prompt tokens, context tokens per round and agent, total token counts) to facilitate a fairer and more practical assessment of MAD relative to single-agent baselines?

---

> ### Author Response · Authors · 2025-11-21
> **Response to Reviewer 2Jxw (1/2)**
>
> Thank you so much for recognizing the strength of our paper and providing thoughtful and insightful feedback. We will address your concerns one by one. We have simplified your comments for easier reference and included our respective responses.
>
> ---
>
> ### **W1: Experiments on Different Domains**
> > The study categorizes tasks into two broad regimes... however, for each category, the authors select only one specific type of task (mathematics and safety, respectively), which limits the representativeness...
>
> **Response:**
> We fully agree that broader task coverage strengthens the generalizability of our framework. To address this, we have significantly expanded our experimental suite in the revised manuscript:
>
> * **Expanded Solution-Finding:** In addition to mathematical reasoning, we added **GPQA-Diamond**, a difficult science benchmark requiring deep reasoning.
> * **Expanded Response-Judgment:** In addition to safety, we added **RewardBench2**, a standard benchmark for evaluating reward models and judgment capabilities.
>
> Our results on these new domains are consistent with our initial findings: heterogeneous MAD provides marginal gains for GPQA (Solution-Finding) but is highly effective for RewardBench (Response-Judgment). This confirms that our taxonomy holds across different domains, not just Math and Safety.
>
> These additional results and analyses are available in ${\color{blue}\text{Appendix G (Page 24)}}$.
>
> ---
>
> ### **W2: Novelty**
> > Prior research ([1]) appears to have explored similar comparative analyses between MAD and Self-Consistency (SC)... yet no citation or discussion of this work is provided in the manuscript.
>
> **Response:**
> We appreciate you pointing this out. We have added a detailed discussion and comparison with [1] in our **Related Work** section (${\color{blue}\text{Appendix A, Page 15}}$) and the **Introduction** (${\color{blue}\text{Section 1, Page 1}}$).
>
> While we share the premise with [1] that MAD is often overvalued compared to baselines like Self-Consistency, our contributions diverge significantly in **depth of analysis** and **final conclusions**:
>
> * **Mechanism vs. Observation:** While [1] observes *that* MAD often underperforms, we explain *why* through the lens of **Test-Time Scaling**. We rigorously decompose the performance gains to isolate whether they stem from *interaction* (Diverse Exploration & Collaborative Refinement) or simply *compute*. This mechanistic decomposition is absent in prior work.
> * **Conditional Effectiveness (Task Nuance):** We move beyond a binary assessment. We identify that MAD is not simply "good" or "bad," but **ineffective for Solution-Finding tasks** (where it acts as compute scaling) yet **highly effective for Response-Judgment tasks** (where interaction adds unique value). We support this with a "Best-of-N" analysis, demonstrating that for math, diversity rarely helps because if the strongest model fails, weaker models rarely succeed.
> * **Contrast in Conclusions:** [1] concludes by recommending that we "embrace model heterogeneity." Our findings refine this significantly: we show that **heterogeneity is not a universal solution**. It fails to improve performance in Solution-Finding tasks (like Math) but is the critical factor for success in Response-Judgment tasks.
>
>
> ---
>
> ### **W3: Unreliable Judge**
> > The reliance solely on GPT-4o-mini for evaluating the Attack Success Rate (ASR) in safety tasks raises concerns... incorporating human validation or cross-validation with more robust evaluators would significantly enhance confidence...
>
> **Response:**
> To verify the reliability of our evaluation, we conducted cross-validation experiments using **Gemini-2.5-Pro**, a significantly stronger model than GPT-4o-mini.
>
> The results using the stronger judge show the exact same trends as our original evaluation. This confirms that the observed effectiveness of MAD in safety tasks is a robust phenomenon and not an artifact of a weak evaluator.
>
> These validation results are included in ${\color{blue}\text{Appendix H.5}}$.
>
> ---
>
> ### **W4: Fair Comparison on Efficiency**
> > ...MAD methods incur additional overhead in context tokens due to cross-agent interactions and extended prompts. This aspect is not quantified...
>
> **Response:**
>
>
> As explained in Appendix E.3, we ensure only that all methods generate the same number of responses, following the setup in the **original test-time scaling paper** [2]. For a more detailed analysis: assuming sufficient memory, methods that require more rounds will be slower, while parallel execution remains the fastest. If instead we assume memory is limited such that only one generation can be processed at a time, then all methods incur similar costs because responses are generated sequentially.
>
> MAD does introduce additional context tokens, but the overhead from input context is relatively small compared to generation cost. This is also consistent with widely observed differences between input and output token costs for API models.

---

> ### Author Response · Authors · 2025-11-21
> **Response to Reviewer 2Jxw (2/2)**
>
> ---
>
> ### **W5: Typo**
> > ...revised the typo “ ASR” -> “Accuracy” in figure 9.
>
> **Response:**
> Thank you for your detailed observation. We have corrected the label from "ASR" to "Accuracy" in Figure 9 in the updated manuscript.
>
> ---
>
> **References**
>
> [1] *Stop Overvaluing Multi-Agent Debate -- We Must Rethink Evaluation and Embrace Model Heterogeneity* \
> [2] *Scaling LLM Test-Time Compute Optimally can be More Effective than Scaling Model Parameters*

---

> ### Author Response · Authors · 2025-11-27
>
> Dear Reviewer 2Jxw,
>
> We truly appreciate your time and effort in reviewing our work.
>
> As the discussion period is nearing its end, we kindly remind you that only a few days remain for further comments or questions.
>
> In response to your feedback, we have provided a detailed response and added a summary of our response recently.
>
> We kindly ask if you have any additional concerns or questions that we may address during the remaining discussion period.
>
> Thank you once again for your valuable insights.
>
> Best regards,
> The Authors

---

### Official Review · Reviewer_7HNX · 2025-10-31

**Soundness:** 2
**Presentation:** 2
**Contribution:** 2
**Rating:** 4
**Confidence:** 3

**Summary:**

This paper reframes multi-agent debate (MAD) as a form of test-time scaling that mixes parallel exploration and sequential collaborative refinement, and asks when (and for what tasks) MAD actually helps over strong single-agent baselines such as self-consistency (SC) and self-refinement (SR). Through controlled comparisons on mathematical reasoning (MATH500, AIME 2024/2025) and safety evaluations (Anthropic Harmful, MultiJail), the authors find that homogeneous MAD rarely beats parallel SC for math; gains appear only on the hardest problems where collaborative refinement can surface and verify a rare correct trajectory. Adding diversity via personas/prompting yields small additional math gains, while mixing different model families tends to regress toward a harmonic-mean–like performance and may destabilize consensus, limiting benefits versus simply sampling more from the strongest model. In contrast, for safety tasks, collaborative refinement with more (and larger) agents consistently reduces attack success rates, and heterogeneous configurations (different prompts or model families) are especially effective. The study’s takeaways are practical: prefer SC for math unless problems are extremely difficult (where MAD can verify a rare correct path), and use heterogeneous MAD for safety to counter bias propagation and improve robustness.

**Strengths:**

* This paper focuses on how and when to choose appropriate design decision for MAD systems, which is a critical challenge in the application of MAD.
* The paper is well written with highlighted takeaways, offering practical guidance to MAD design.

**Weaknesses:**

* The idea of agent diversity, and the observation that MAD sometimes fails single agent methods have been well acknowledged in previous research [1][2][3]
* While with comprehensive evaluation and practical design recommendations, this paper did not offer a unified solution to this problem, limiting the technical contribution.
* The takeaways are bounded by task types, difficulty, which is sensitive to the selection of base models, and may not be generalizable to stronger models or stronger reasoning models, especialy given that MADs are naturally sensitive to prompts. It is not promising that these takeaways can benefit further MAD system design.


[1] Reconcile: Round-table conference improves reasoning via consensus among diverse llms
[2] Stop Overvaluing Multi-Agent Debate -- We Must Rethink Evaluation and Embrace Model Heterogeneity
[3] Should we be going MAD? A Look at Multi-Agent Debate Strategies for LLMs

**Questions:**

See weaknesses.

---

> ### Author Response · Authors · 2025-11-21
> **Response to Reviewer 7HNX**
>
> Thank you for your feedback. We will address your concerns one by one. We have simplified your comments for easier reference and included our respective responses.
>
> ---
>
> ### **W1: On Novelty**
>
> > The idea of agent diversity, and the observation that MAD sometimes fails single agent methods have been well acknowledged in previous research [1][2][3].
>
> **Response:**
> There are indeed works showing that “MAD sometimes fails compared to single-agent methods,” but there are also works demonstrating that MAD can be effective. This mixed evidence motivates a more in-depth investigation, as noted in the Related Work section. Our work moves significantly beyond these initial observations by providing a **systematic explanatory framework**.
>
> * **Beyond "Observation" to "Mechanism":** While previous papers observe *that* MAD sometimes fails, they do not rigorously explain *why* or distinctively isolate the source of the gain (interaction vs. compute). We provide this by defining MAD as a **Test-Time Scaling** framework.
> * **Identifying the Boundary Conditions:** We do not simply argue that "MAD sometimes fails." Instead, we introduce a clear taxonomy distinguishing **Solution-Finding Tasks** (where MAD fails) from **Response-Judgment Tasks** (where heterogenous MAD suceeds).
> * **Sophisticated Analysis of Diversity:** Unlike [1] or [3], which analyze consensus or general strategies, we quantitatively decompose the role of diversity. We show that for Solution-Finding, even high diversity fails to outperform single-agent scaling if the "Best-of-N" condition isn't met, whereas for Judgment tasks, diversity is the primary driver of success.
>
> We have clarified these distinctions in the Related Work section (${\color{blue}\text{Appendix A, Page 15}}$) and the Introduction (${\color{blue}\text{Section 1, Page 1}}$).
>
> ---
>
> ### **W2: No Unified Solution**
>
> > ...this paper did not offer a unified solution to this problem, limiting the technical contribution.
>
> **Response:**
> We respectfully suggest that providing a single "silver bullet" algorithm is not the only form of technical contribution. In a field saturated with conflicting results, **establishing rigorous boundary conditions** is equally critical.
>
> * **The "Framework" is the Solution:** Our primary contribution is establishing the **conditional effectiveness** of MAD. The literature currently suffers from oversimplified conclusions (e.g., "MAD is good" vs. "MAD is bad"). Our framework *is* the unified solution to this inconsistency: it provides a decision matrix for practitioners to know *when* to apply MAD based on task characteristics, model scale, and available diversity.
> * **Resolving Inconsistencies:** By demonstrating that efficacy varies significantly based on task type (Solution-Finding vs. Judgment), we resolve the contradictions found in prior studies. A "unified algorithmic solution" that ignores these fundamental task differences would likely be suboptimal.
>
> We have emphasized this contribution in the Discussion section (${\color{blue}\text{Section 6, Page 10}}$).
>
> ---
>
> ### **W3: Generalization with Stronger Models**
>
> > The takeaways are bounded by task types... and may not be generalizable to stronger models... especially given that MADs are naturally sensitive to prompts.
>
> **Response:**
> We understand the concern regarding generalization, but our empirical evidence suggests our findings describe fundamental dynamics rather than artifacts of specific weak models.
>
> * **Consistency Across Scales:** Our experiments were not limited to small models. We evaluated over 5 model variants ranging from small (1.5B) to large (32B) and proprietary models. Crucially, we observed a **consistent trend** across all scales: the dichotomy between Solution-Finding and Response-Judgment tasks holds regardless of model strength.
> * **Fundamental Dynamics vs. Prompt Sensitivity:** While prompts matter, the core limitation we identified—that interaction cannot create knowledge in Solution-Finding tasks if the agents lack the capability—is a fundamental property of the system, not a prompt sensitivity. Conversely, the benefit in Response-Judgment tasks scales reliably with model capability.
>
> Our original manuscript also contains the analysis regarding model scaling (${\color{blue}\text{Appendix H, Page 25-29}}$).
>
> ---
>
> **References** \
> [1] *Reconcile: Round-table conference improves reasoning via consensus among diverse llms* \
> [2] *Stop Overvaluing Multi-Agent Debate -- We Must Rethink Evaluation and Embrace Model Heterogeneity* \
> [3] *Should we be going MAD? A Look at Multi-Agent Debate Strategies for LLMs*

---

> ### Author Response · Authors · 2025-11-27
>
> Dear Reviewer 7HNX,
>
> We truly appreciate your time and effort in reviewing our work.
>
> As the discussion period is nearing its end, we kindly remind you that only a few days remain for further comments or questions.
>
> In response to your feedback, we have provided a detailed response and added a summary of our response recently.
>
> We kindly ask if you have any additional concerns or questions that we may address during the remaining discussion period.
>
> Thank you once again for your valuable insights.
>
> Best regards,
> The Authors

---

### Official Review · Reviewer_LsAr · 2025-11-01

**Soundness:** 3
**Presentation:** 3
**Contribution:** 1
**Rating:** 2
**Confidence:** 4

**Summary:**

This paper systematically evaluates Multi-Agent Debate (MAD) against strong baselines (Self-Consistency and Self-Refinement) on mathematical reasoning and safety tasks. It finds MAD's effectiveness is highly conditional: it offers limited benefits in math, except for very difficult problems when using weaker models , and it increases vulnerability in safety tasks unless heterogeneous agents are employed.

**Strengths:**

This paper is empirically thorough, systematically testing MAD performance across a range of tasks, difficulty levels, and model scales, demonstrating that MAD is not a universally superior method but is beneficial only under specific, well-defined conditions.

**Weaknesses:**

The primary weakness is a severe lack of novelty. The paper's core findings are nearly identical to those in a recent, highly related publication If multi-agent debate is the answer, what is the question? (arXiv:2502.08788). Both perform a systematic evaluation of MAD and arrive at the same two conclusions:

1.MAD is Overvalued: Both papers find that MAD generally fails to outperform strong single-agent baselines like Self-Consistency (SC).

2.Heterogeneity is the Solution: Both papers identify model heterogeneity (using diverse agents) as the key solution.

The prior work already established heterogeneity as the 'universal antidote'. This paper's primary conclusions are therefore a re-discovery, which significantly limits its contribution. The authors must address this major overlap.

**Questions:**

Please respond to the above weakness.

---

> ### Author Response · Authors · 2025-11-21
> **Response to Reviewer LsAr**
>
> ### **W1: Similar Observation on MAD (MAD is overvalued)**
>
> > The paper's finding that MAD is often overvalued compared to single-agent baselines seems identical to the findings in [1].
>
> **Response:**
> We acknowledge that both our work and [1] share a starting premise: that the community’s enthusiasm for MAD requires critical re-evaluation. However, as mentioned in our general response, our novelty lies in **(1) defining how MAD fundamentally differs from single-agent test-time scaling** and **(2) determining how these features drive conditional effectiveness and explaining the underlying mechanisms.**
>
> * **MAD as a Test-Time Scaling Framework:** As detailed in our **General Response**, we analyze MAD through the lens of **Test-Time Scaling**. We isolate whether the gain comes from *interaction* or *compute*. We achieve this by defining the key features of MAD—*Diverse Exploration* and *Collaborative Refinement*—and isolating how these affect performance across different axes: compute scaling, task types, and agent diversity.
> * **Beyond "Overvalued":** While [1] primarily argues that MAD is overvalued because single-agent baselines (like Self-Consistency) can match it, our work goes further to define the **conditional boundaries** of MAD. We do not simply conclude that "MAD is bad/overvalued." Instead, we identify that MAD is:
>     1.  **Ineffective** for **Solution-Finding** tasks (e.g., Math, Logic), where it acts merely as compute scaling.
>     2.  **Highly Effective** for **Response-Judgment** tasks (e.g., Safety, RewardBench), where the interaction dynamics provide value beyond simple compute.
>
> We also provide a deep analysis of *why* this occurs. For example, in mathematical reasoning, we investigate why agent diversity does not help. Our data shows that there are very few instances where the best-performing model fails but weaker models succeed. This "Best-of-N" analysis of heterogeneous agents is critical to understanding the limitations of the MAD framework.
>
> We have explicitly added this detailed comparison in the Related Work section (${\color{blue}\text{Appendix A, Page 15}}$) and clarified the distinction in the Introduction (${\color{blue}\text{Section 1, Page 1}}$).
>
> ---
>
> ### **W2: Heterogeneity is the Solution**
>
> > The conclusion that model heterogeneity improves performance is also a core finding of [1].
>
> **Response:**
> While [1] concludes with a broad recommendation to "embrace heterogeneity," our findings regarding heterogeneity are strictly **conditional** and offer a significant counter-point to [1].
>
> * **Nuanced Conclusion vs. Blanket Recommendation:** [1] suggests heterogeneity is generally the solution to MAD's stagnation. In contrast, we empirically show that **heterogeneity is NOT a solution** for **Solution-Finding** tasks.
>     * In our experiments (including the newly added GPQA-Diamond), even when using highly heterogeneous agents, MAD does not significantly outperform single-agent scaling for reasoning tasks with objective answers.
>     * Heterogeneity becomes the primary driver of performance **only** in **Response-Judgment** tasks.
> * **Mechanism of Action:** We explain *why* this happens. In Solution-Finding, if agents lack the capability to solve the core problem, diverse perspectives do not help them converge on the truth. In Judgment tasks, diverse perspectives help cover the search space of potential flaws/biases.
>
> This distinction—that heterogeneity is not a universal fix—is a key novelty of our paper compared to [1]. We have clarified this in our updated Discussions (${\color{blue}\text{Section 6, Page 10}}$).
>
> ---
>
> **References**
>
> [1] *Stop Overvaluing Multi-Agent Debate -- We Must Rethink Evaluation and Embrace Model Heterogeneity*

---

> ### Author Response · Authors · 2025-11-27
>
> Dear Reviewer LsAr,
>
> We truly appreciate your time and effort in reviewing our work.
>
> As the discussion period is nearing its end, we kindly remind you that only a few days remain for further comments or questions.
>
> In response to your feedback, we have provided a detailed response and added a summary of our response recently.
>
> We kindly ask if you have any additional concerns or questions that we may address during the remaining discussion period.
>
> Thank you once again for your valuable insights.
>
> Best regards,
> The Authors

---

### Author Response · Authors · 2025-11-21
**General Response 2/2 (Novelty, Experiments, and Clarification)**

### **Responses to Common Concerns**

#### **W1: Novelty and Distinction from Prior Work**
*Common Question: How does this work differ from previous studies like [1], which also compared MAD against single-agent baselines?*

**Response:**
While both our work and [1] perform systematic comparisons, our framework and core findings diverge significantly. Our objective is not to issue a binary verdict on whether MAD is "better," but to uncover **"when, why, and how"** MAD provides value. This is more practical and provides better insights, as it guides the user on  *which tasks to apply MAD to*, rather than *simply whether to apply MAD or no*t. We also include in-depth analysis explaining why for each question. Specifically, our contributions are as follows:

1.  **Test-Time Scaling Perspective:** We are the first to rigorously define MAD through the lens of **test-time scaling**. We isolate whether performance gains stem from the specific interactions inherent in MAD—diverse exploration and collaborative refinement—or simply from increased compute.
2.  **Conditional Effectiveness:** Our insights highlight that the **nature of the task** is the primary determinant of success.
    * For **Solution-Finding** tasks (e.g., general reasoning), we find that MAD offers very marginal gains, even with heterogeneous agents.
    * For **Response-Judgment** tasks, MAD yields significant benefits, provided agent diversity is maintained.
3.  **Nuanced Conclusion:** Unlike previous works that often conclude either "MAD is generally good" or "Heterogeneous MAD is good while Homogeneous is bad," our work provides a granular framework showing that effectiveness is highly conditional on the task type, further supported by our analysis across varying diversity levels and model scales.


We have revised the Introduction (${\color{blue}\text{Page 1}}$) and Related Work (${\color{blue}\text{Appendix A, Page 15}}$) to explicitly articulate these distinctions.

#### **W2: Limited Task Scope**
*Common Question: The initial draft focused primarily on mathematical reasoning and safety. Does the framework hold for other domains?*

**Response:**
We agree that broader task coverage strengthens our claims. To address this, we have expanded our experimental suite to cover a wider spectrum of task families.

1.  **New Benchmarks:** We added **GPQA-Diamond** to represent complex *Solution-Finding* tasks and **RewardBench2** to represent *Response-Judgment* tasks.
2.  **Consistent Findings:** Our results on these new tasks align perfectly with our initial hypothesis:
    * On **GPQA (Solution-Finding)**, MAD does not bring clear gains compared to single-agent scaling.
    * On **RewardBench (Response-Judgment)**, MAD is highly effective.
3.  **Implication:** This confirms that MAD is most effective in subjective domains or tasks with limited search spaces (judging), rather than in open-ended solution spaces with gold-standard answers.

The detailed results and analysis for these additional tasks are provided in ${\color{blue}\text{Appendix G (Page 24)}}$.

---

**Detailed Responses**
In addition to this general response, we have provided detailed point-by-point responses to specific questions within the **Official Comment** for each reviewer.

Thank you again for your constructive feedback. We believe these updates significantly strengthen the manuscript. We are happy to engage in further discussion if there are any remaining questions.

**References:**

[1] *Stop Overvaluing Multi-Agent Debate -- We Must Rethink Evaluation and Embrace Model Heterogeneity*

---

### Author Response · Authors · 2025-11-21
**General Response 1/2 (PDF Updates)**

We thank the reviewers for their time and valuable feedback. We appreciate that the reviewers recognized our work as empirically thorough, rigorous, and well-written. We are also grateful for the acknowledgement of our analysis regarding the internal dynamics that explain the **conditional effectiveness** of Multi-Agent Debate (MAD).

In this rebuttal, we have focused on clarifying our **novelty** and contributions, justifying our **experimental design**, and providing **additional experiments** to strengthen our claims. Below, we outline the specific updates made to the manuscript, followed by responses to the common concerns raised.

---

### **Summary of Updates in the Revised PDF**

We have revised the manuscript to address the reviewers' comments. All updated sections in the revised PDF are marked in ${\color{blue}\text{blue}}$.

* **Clarifying Novelty & Contributions:**
    * We have explicitly included a comparison with [1] and other relevant works to emphasize our unique contributions. This discussion is added to the **Related Work** section (${\color{blue}\text{Appendix A, Page 15}}$).
    * We revised the introduction to clearly frame our primary goal: Defining the unique features of MAD as a form of test-time scaling and identifying the factors that determine its conditional effectiveness (${\color{blue}\text{Section 1, Page 1}}$). We move beyond simply asking "is MAD good?" to answering "when and why is it effective?", with a specific focus on task types.

* **Expansion of Task Families:**
    * To emphasize our novelty and address concerns about evaluation on only two tasks, we formally distinguish between **Solution-Finding Tasks** and **Response-Judgment Tasks**. This distinction is clarified in the Introduction (${\color{blue}\text{Section1, Page 2}}$) and Experimental Setting (${\color{blue}\text{Section 3, Page 4}}$).
    * We have added results for **two new benchmarks** to strengthen our analysis:
        1.  **GPQA-Diamond** (representing Solution-Finding).
        2.  **RewardBench2** (representing Response-Judgment).
        * These results and the accompanying analysis are included in ${\color{blue}\text{Appendix G, Page 24)}}$.

* **Judge Reliability Experiments:**
    * To ensure the robustness of our evaluation, we conducted additional experiments on safety tasks using a stronger judge model (*Gemini-2.5-Pro*). The results demonstrate the reliability of our evaluation framework and are available in ${\color{blue}\text{Appendix H.5, Page 29}}$.

* **Writing & Clarification:**
    * We have added necessary details regarding our evaluation protocols (${\color{blue}\text{Section 3, Page 4}}$).
    * We have unified our analysis into a cohesive conclusion (${\color{blue}\text{Section 6, Page 10}}$).
    * We have corrected all typos and grammatical errors pointed out by the reviewers.

---

### Meta-Review · Area_Chair_rMqV · 2026-01-06

**Summary:**

This paper reframes Multi-Agent Debate (MAD) as a form of test-time scaling and systematically studies when MAD helps compared to strong single-agent baselines such as self-consistency and self-refinement. Reviewers generally agreed that the paper is clearly written, empirically thorough, and offers careful analysis of internal MAD dynamics. However, the dominant concerns focused on limited novelty relative to very recent closely related work, the narrowness of the evaluated task families, and uncertainty about the broader significance and generalizability of the conclusions.

**Reviewer Concerns:**

In the rebuttal, the authors made a genuine effort to clarify their novelty, reorganize the message around conditional effectiveness, and added new benchmarks to strengthen the empirical scope. These changes help improve clarity and reinforce the internal consistency of the study. However, the most critical concerns are only partially mitigated. In particular, multiple reviewers emphasized strong overlap with very recent work that already reports similar conclusions about MAD and heterogeneity, and this novelty gap is not fully resolved by reframing alone. In addition, although extra benchmarks were added, the evaluation still centers on a limited set of task types, and several reviewers remained unconvinced that the proposed takeaways generalize beyond the chosen math and safety settings. Questions about whether the conclusions meaningfully advance MAD system design, rather than consolidating existing observations, therefore remain.

**Reviewer Scores:**

No reviewer explicitly signaled a positive score change after rebuttal. The initial scores clustered around 2 and several 4s, with concerns concentrated on contribution and significance rather than missing details. I believe most reviewers would likely maintain their original scores after full discussion, as the rebuttal improves presentation and empirical support but does not fundamentally shift the novelty and generalization assessment.

---

### Decision · Program_Chairs · 2026-01-26

Reject